# A Generalization of the Bivariate Gamma Distribution Based on Generalized Hypergeometric Functions

Christian Caamaño-Carrillo [1] and Javier E. Contreras-Reyes [2,*]

1 Departamento de Estadística, Facultad de Ciencias, Universidad del Bío-Bío, Concepción 4081112, Chile; chcaaman@ubiobio.cl
2 Instituto de Estadística, Facultad de Ciencias, Universidad de Valparaíso, Valparaíso 2360102, Chile
* Correspondence: jecontrr@uc.cl

**Abstract:** In this paper, we provide a new bivariate distribution obtained from a Kibble-type bivariate gamma distribution. The stochastic representation was obtained by the sum of a Kibble-type bivariate random vector and a bivariate random vector built by two independent gamma random variables. In addition, the resulting bivariate density considers an infinite series of products of two confluent hypergeometric functions. In particular, we derive the probability and cumulative distribution functions, the moment generation and characteristic functions, the Hazard, Bonferroni and Lorenz functions, and an approximation for the differential entropy and mutual information index. Numerical examples showed the behavior of exact and approximated expressions.

**Keywords:** generalized gamma distribution; generalized hypergeometric function; moment generation function; differential entropy; mutual information

**MSC:** 62E17; 33B20; 94A17



## 1. Introduction

Let $Z_{ik}$, $i = 1, \ldots, \nu$, $\nu > 2$, an independent sequence of standardized normal random variables with correlation $Corr(Z_{ik}, Z_{ik}) = \rho$, $i \neq j$, and let $U_k = \sum_{i=1}^{\nu} Z_{ik}^2$ with $k = 1, 2$, then $U_k$ is a random variable with $\chi_\nu^2$ marginal distribution. According to [1], the distribution of $\mathbf{U} = (U_1, U_2)^\top$ has a correlated bivariate chi-square distribution with $\nu$ degrees of freedom parameter and probability density function (pdf) given by

$$f_{\mathbf{U}}(\mathbf{u}) = \frac{2^{-\nu}(u_1 u_2)^{(\nu-2)/2} e^{-\frac{(u_1+u_2)}{2(1-\rho^2)}}}{\Gamma^2\left(\frac{\nu}{2}\right)(1-\rho^2)^{\nu/2}} F_{0,1}\left(\frac{\nu}{2}; \frac{\rho^2 u_1 u_2}{4(1-\rho^2)^2}\right), \tag{1}$$

where $F_{p,q}(a_1, a_2, \ldots, a_p; b_1, b_2, \ldots, b_p; x)$ denotes the generalized hypergeometric function defined by

$$F_{p,q}(a_1, a_2, \ldots, a_p; b_1, b_2, \ldots, b_q; x) = \sum_{k=0}^{\infty} \frac{(a_1)_k (a_2)_k \cdots (a_p)_k}{(b_1)_k (b_2)_k \cdots (b_q)_k} \frac{x^k}{k!}, \tag{2}$$

for $p, q = 0, 1, 2, \ldots$, and with $(a)_k = \frac{\Gamma(k+a)}{\Gamma(a)}$, $k \in \mathbb{N} \cup \{0\}$, being the Pochhammer symbol [2]. Note that in this case, the correlation function of $U$ is given by $\rho_U = \rho^2$.

Transformation $W = \frac{U}{\beta}$, $\beta > 0$ defines a random variable $Gamma(\nu/2, \beta/2)$ distributed with density

$$f_W(w) = \left(\frac{\beta}{2}\right)^{\nu/2} \frac{w^{\nu/2-1}}{\Gamma\left(\frac{\nu}{2}\right)} e^{-\frac{\beta}{2}w}, \quad w > 0. \tag{3}$$

The pdf of $\mathbf{W} = (W_1, W_2)^\top$ has a Kibble-type correlated bivariate gamma distribution [3], given by

$$f_{\mathbf{W}}(\mathbf{w}) = \frac{2^{-\nu} \beta^\nu (w_1 w_2)^{\nu/2-1} e^{-\frac{\beta}{2(1-\rho^2)}(w_1+w_2)}}{\Gamma\left(\frac{\nu}{2}\right)^2 (1-\rho^2)^{\nu/2}} F_{0,1}\left(\frac{\nu}{2}; \frac{\beta^2 \rho^2 w_1 w_2}{4(1-\rho^2)^2}\right), \tag{4}$$

where $\mathbb{E}(W) = \frac{\nu}{\beta}$, $Var(W) = \frac{2\nu}{\beta^2}$ and $\rho_W = \rho^2$. In Equations (1) and (4), the case $\rho = 0$ implies the product of two independent chi-square and gamma random variables (i.e., the same result obtained by the bivariate normal distribution in the independency case).

The univariate and bivariate gamma distributions are basic distributions that have been used to model data in many applications [4,5]. More recently, several examples of bivariate distributions and their applications emerged: streamflow data [6], drought data modeling [7], rainfall data modeling [8], wind speed data spatio-temporal modeling [9], flood volume-peak data modeling [10], wireless communications models [11], and transmit antennas system modeling [12].

In this paper, we build a generalization of bivariate gamma distribution using a Kibble type bivariate gamma distribution. The stochastic representation was obtained by the sum of a Kibble-type bivariate random vector and a bivariate random vector built by two independent gamma random variables. In addition, the resulting bivariate density considers an infinite series of products of two confluent hypergeometric functions. In particular, we derive the pdf and cumulative distribution function (cdf), moment generation and characteristic functions, Hazard, Bonferroni and Lorenz functions, and an approximation for the differential entropy and mutual information index. Numerical examples showed the behavior of exact and approximated expressions. All numerical examples were calculated using the `hypergeo` package of `R` software [13].

The paper is organized as follows. Section 2 presents the generalization of the bivariate gamma distribution, with its pdf (with simulations), cdf, moment generation and characteristic functions, cross-product moment, covariance and correlation (with simulations), and some special expected values. Moreover, the Hazard, Bonferroni and Lorenz functions are computed. Section 3 presents the approximation for the differential entropy and mutual information index (with simulations) for the generalized bivariate gamma distribution with some numerical results. The paper ends with a discussion in Section 4. Proofs are available in Appendix A section.

## 2. Bivariate Gamma Generalization

Let

$$Y = W + R, \tag{5}$$

where $R$ is a random variable, $R \sim Gamma(\alpha/2, \beta/2)$, $\alpha > 0$ and the distribution of $W$ is defined by Equation (3); thus $Y \sim Gamma((\alpha+\nu)/2, \beta/2)$ is a random variable with marginal gamma distribution and arbitrary shape, $(\alpha+\nu)/2$, and scale, $\beta/2$, based on $\alpha$, $\beta$ and $\nu$ parameters. This type of construction has been proposed by [5,6,14–19] to build a bivariate gamma distribution. Specifically, the authors considered the case $W_1 = W_2$ in (3). Properties of the bivariate gamma distribution can be found in [15,18,20,21].

In line with the stochastic representation (5), we consider the bivariate distribution of $\mathbf{Y} = (Y_1, Y_2)^\top$ as a generalization of the Cheriyan distribution [15], where

$$Y_1 = W_1 + R_1,$$
$$Y_2 = W_2 + R_2,$$

$\mathbf{W} = (W_1, W_2)^\top$ is given in (3) and (4) with $\rho_W = \rho^2$, and $R_i \sim Gamma(\alpha_i/2, \beta/2)$, $\alpha_i > 0$, $R_i \perp R_j, \forall i \neq j$, $R_i \perp W_j, \forall i, j$. Thus, $Y \sim Gamma((\alpha_k + \nu)/2, \beta/2)$, $k = 1, 2$.

In the following theorem, we provide a new bivariate distribution with gamma marginal distributions obtained using a Kibble-type bivariate gamma distribution [3].

**Theorem 1.** *Let $W_k = \sum_{i=1}^{\nu} Z_{ik}^2/\beta$ where $Z_{ik}$, $i = 1, \ldots, \nu$, $k = 1, 2$, is a finite sequence of independent normal random variables with zero mean, unit variance, and correlation $\rho$. Let $\mathbf{Y} = \mathbf{W} + \mathbf{R}$, with $\mathbf{W} = (W_1, W_2)^\top$ and $R_k \sim Gamma(\alpha_k/2, \beta/2)$, $k = 1, 2$. The pdf of $\mathbf{Y} = (Y_1, Y_2)^\top$ is given by*

$$f_{\mathbf{Y}}(\mathbf{y}) = \frac{\left(\frac{\beta}{2}\right)^{\nu + \frac{\alpha_1 + \alpha_2}{2}} y_1^{(\nu + \alpha_1)/2 - 1} y_2^{(\nu + \alpha_2)/2 - 1} e^{-\frac{\beta}{2(1-\rho^2)}(y_1 + y_2)}}{\Gamma\left(\frac{\nu + \alpha_1}{2}\right) \Gamma\left(\frac{\nu + \alpha_2}{2}\right) (1 - \rho^2)^{\nu/2}} \sum_{k=0}^{\infty} \frac{\left(\frac{\nu}{2}\right)_k}{k! \left(\frac{\nu + \alpha_1}{2}\right)_k \left(\frac{\nu + \alpha_2}{2}\right)_k} \left(\frac{\beta^2 \rho^2 y_1 y_2}{4(1 - \rho^2)^2}\right)^k$$

$$\times F_{1,1}\left(\frac{\alpha_1}{2}; \frac{\nu + \alpha_1}{2} + k; \frac{\beta \rho^2 y_1}{2(1 - \rho^2)}\right) F_{1,1}\left(\frac{\alpha_2}{2}; \frac{\nu + \alpha_2}{2} + k; \frac{\beta \rho^2 y_2}{2(1 - \rho^2)}\right), \tag{6}$$

*where $F_{1,1}(a_1; b_1; x)$ is the confluent hypergeometric function defined in (2) for $p = q = 1$.*

Theorem 1 shows that the pdf considers an infinite series of products of two confluent hypergeometric functions. Note that when $\rho = 0$, pdf in Theorem 1 becomes the product of two independent gamma random variables, $Gamma((\nu + \alpha_k)/2, \beta/2)$, $k = 1, 2$, i.e., the same property of the bivariate normal distribution is accomplished. When $\alpha_1 = \alpha_2 = 0$, then $f_{\mathbf{Y}}(\mathbf{y}) = f_{\mathbf{W}}(\mathbf{w})$, i.e., $\mathbf{Y}$ is Kibble-type gamma distributed (see Figure 1).

Figure 1 shows the pdf of Equation (6) for some parameters of $\mathbf{Y}$. When $\rho$ increases, is produced the largest values of $y_1$ and $y_2$ in the pdf. When $\rho = 0.25$, the pdf is close at origin $(y_1, y_2 \approx 0)$, has positive bias and decays exponentially. When $\rho = 0.5$, the pdf has less bias, but more symmetry and variability. When $\rho = 0.75$, the pdf has a bias to the right, but with less bias than in the case $\rho = 0.25$. When $\beta$ increases (decreases), its variance increases (decreases). When $\nu$ increases, the pdf shows heavy-tailed behavior, at the same time as parameter $\alpha_k$, $k = 1, 2$ increases (as the usual gamma distribution).

**Theorem 2.** *The joint cdf of $\mathbf{Y} = (Y_1, Y_2)^\top$ in Equation (6) can be expressed as*

$$F_{\mathbf{Y}}(Y_1 \leq t_1, Y_2 \leq t_2) = (1 - \rho^2)^{\frac{\nu + \alpha_1 + \alpha_2}{2}} \sum_{k=0}^{\infty} \frac{\left(\frac{\nu}{2}\right)_k \rho^{2k}}{k!} \gamma_{2,1}\left(\frac{\alpha_1}{2}, \left(\frac{\nu + \alpha_1}{2} + k, \frac{\beta t_1}{2(1 - \rho^2)}\right); \frac{\nu + \alpha_1}{2} + k; \rho^2\right)$$

$$\times \gamma_{2,1}\left(\frac{\alpha_2}{2}, \left(\frac{\nu + \alpha_2}{2} + k, \frac{\beta t_2}{2(1 - \rho^2)}\right); \frac{\nu + \alpha_2}{2} + k; \rho^2\right), \tag{7}$$

*where $\gamma_{2,1}(a_1, (a_2, x); b_1; z)$ denotes the incomplete gaussian hypergeometric function as*

$$\gamma_{2,1}(a_1, (a_2, x); b_1; z) = \sum_{k=0}^{\infty} \frac{(a_1)_k (a_2; x)_k}{(b_1)_k} \frac{z^k}{k!},$$

*with incomplete Pochhammer symbols given by $(a_2; x)_k = \frac{\gamma(a_2 + k, x)}{\Gamma(a_2)}$ for $a_2, k \in \mathbb{C}$, $x \geq 0$ [2,22].*

Theorem 2 shows that the joint pdf considers an infinite series of products of two incomplete gaussian hypergeometric functions.

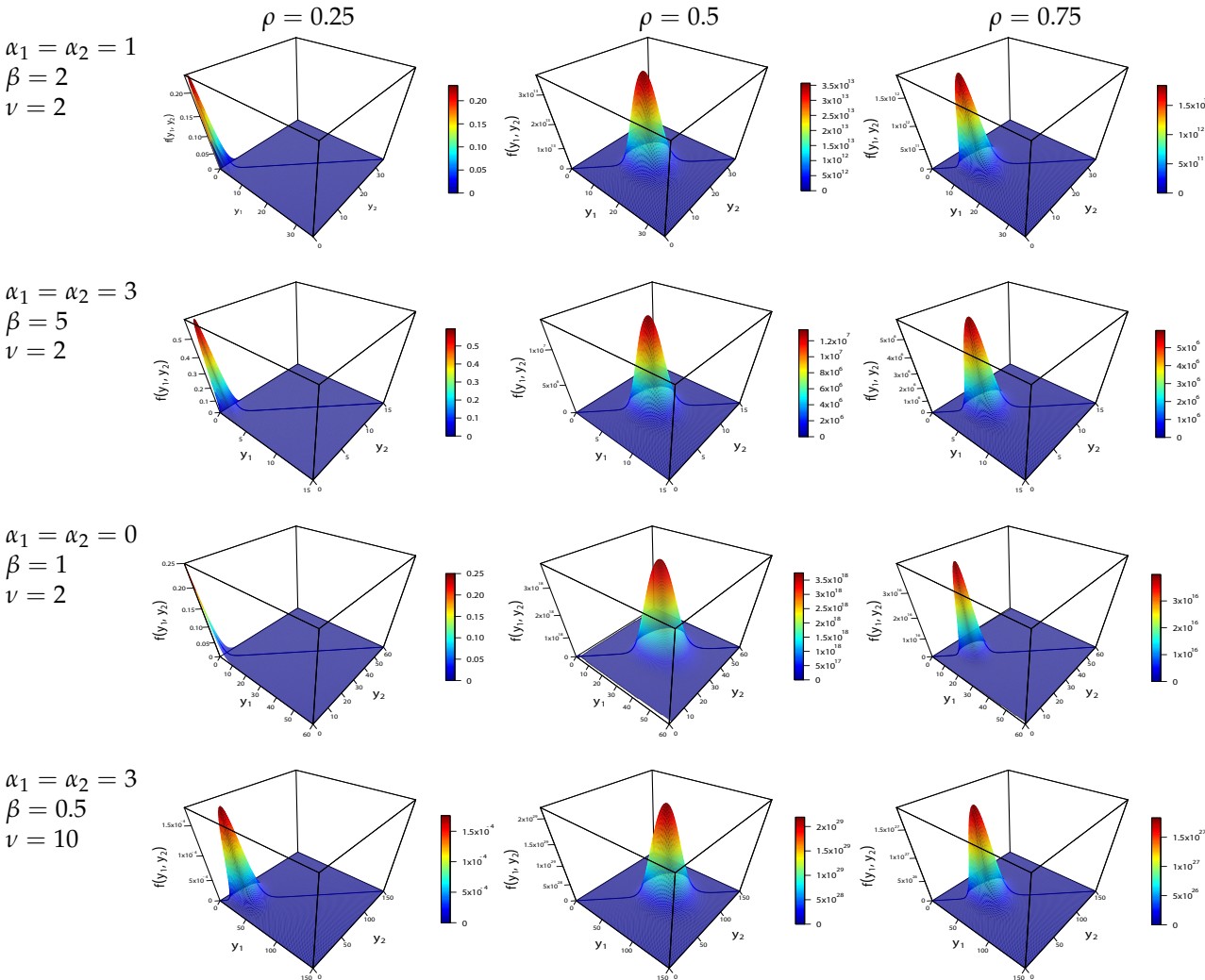

**Figure 1.** Bivariate pdf of Equation (6) for some parameter combinations.

### 2.1. Moment Generation Function

In this section, we analyze the moment generation function of $\mathbf{Y} = (Y_1, Y_2)^\top$, the cross-product moment between $Y_1$ and $Y_2$, and some particular expected values involving these variables.

**Proposition 1.** *The joint moment generation (mgf) and characteristic functions of $\mathbf{Y} = (Y_1, Y_2)^\top$ given in Equation (6) are*

$$M(t_1, t_2) = \beta^{\nu/2} \left( \frac{\beta(1-\rho^2)}{[\beta - 2(1-\rho^2)t_1][\beta - 2(1-\rho^2)t_2] - \beta^2\rho^2} \right)^{\frac{\nu}{2}} \left( \frac{\beta(1-\rho^2)}{(\beta - 2(1-\rho^2)t_1) - \beta\rho^2} \right)^{\frac{\alpha_1}{2}} \left( \frac{\beta(1-\rho^2)}{(\beta - 2(1-\rho^2)t_2) - \beta\rho^2} \right)^{\frac{\alpha_2}{2}} \quad (8)$$

*and*

$$\phi(t_1, t_2) = \beta^{\nu/2} \left( \frac{\beta(1-\rho^2)}{[\beta - 2(1-\rho^2)it_1][\beta - 2(1-\rho^2)it_2] - \beta^2\rho^2} \right)^{\frac{\nu}{2}} \left( \frac{\beta(1-\rho^2)}{(\beta - 2(1-\rho^2)it_1) - \beta\rho^2} \right)^{\frac{\alpha_1}{2}} \left( \frac{\beta(1-\rho^2)}{(\beta - 2(1-\rho^2)it_2) - \beta\rho^2} \right)^{\frac{\alpha_2}{2}}, \quad (9)$$

*respectively.*

Assuming $\rho = 0$, thus $M(t_1, 0) = \left(\frac{\beta}{\beta - 2t_1}\right)^{\frac{\nu + \alpha_1}{2}}$ and $M(0, t_2) = \left(\frac{\beta}{\beta - 2t_2}\right)^{\frac{\nu + \alpha_2}{2}}$, which are the mgf's of a gamma random variable. The proof of the characteristic function of $\mathbf{Y}$ is trivial, following the proof of Proposition 1 for $M(t_1, t_2)$.

**Proposition 2.** *The cross-product moment of* $\mathbf{Y} = (Y_1, Y_2)^\top$ *in Equation (6) can be expressed as*

$$\mathbb{E}(Y_1^a Y_2^b) = \frac{\left(\frac{2}{\beta}\right)^{a+b}(1-\rho^2)^{(\nu+\alpha_1+\alpha_2)/2+a+b}\Gamma\left(\frac{\nu+\alpha_1}{2}+a\right)\Gamma\left(\frac{\nu+\alpha_2}{2}+b\right)}{\Gamma\left(\frac{\nu+\alpha_1}{2}\right)\Gamma\left(\frac{\nu+\alpha_2}{2}\right)} \sum_{k=0}^{\infty} \frac{\left(\frac{\nu}{2}\right)_k \left(\frac{\nu+\alpha_1}{2}+a\right)_k \left(\frac{\nu+\alpha_2}{2}+b\right)_k}{k!\left(\frac{\nu+\alpha_1}{2}\right)_k \left(\frac{\nu+\alpha_2}{2}\right)_k}\rho^{2k}$$

$$\times F_{2,1}\left(\frac{\alpha_1}{2}, \frac{\nu+\alpha_1}{2}+a+k; \frac{\nu+\alpha_1}{2}+k; \rho^2\right) F_{2,1}\left(\frac{\alpha_2}{2}, \frac{\nu+\alpha_2}{2}+b+k; \frac{\nu+\alpha_2}{2}+k; \rho^2\right), \tag{10}$$

*where* $F_{2,1}(a_1, a_2; b_1; x)$ *is the gaussian hypergeometric function defined in (2) for* $p = 2$ *and* $q = 1$.

Proposition 2 shows that the cross-product moment considers an infinite series of products of two gaussian hypergeometric functions. A direct result of Proposition 2 is the following Corollary 1, that presents the expected value and variance of marginal gamma random variable $Y_i$, and the covariance and correlation between two marginal gamma random variables, $Y_1$ and $Y_2$.

**Corollary 1.** *If* $\mathbf{Y} = (Y_1, Y_2)^\top$, *it has pdf according to (6). According to Proposition 2, we have*

(a)    $\mathbb{E}(Y_k) = \frac{\nu + \alpha_k}{\beta}$, $k = 1, 2$.

(b)    $Var(Y_k) = \frac{2(\nu + \alpha_k)}{\beta^2}$, $k = 1, 2$.

(c)

$$Cov(Y_1, Y_2) = \frac{(\nu + \alpha_1)(\nu + \alpha_2)}{\beta^2}\left[(1-\rho^2)^{\nu/2} \sum_{k=0}^{\infty} \frac{\left(\frac{\nu}{2}\right)_k \left(\frac{\nu+\alpha_1}{2}+1\right)_k \left(\frac{\nu+\alpha_2}{2}+1\right)_k}{k!\left(\frac{\nu+\alpha_1}{2}\right)_k \left(\frac{\nu+\alpha_2}{2}\right)_k}\rho^{2k}\right.$$

$$\left. \times \left(1 - \rho^2 \frac{\nu + 2k}{\nu + \alpha_1 + 2k}\right)\left(1 - \rho^2 \frac{\nu + 2k}{\nu + \alpha_2 + 2k}\right) - 1\right].$$

(d)

$$\rho_{\mathbf{Y}} = \frac{\sqrt{(\nu + \alpha_1)(\nu + \alpha_2)}}{2}\left[(1-\rho^2)^{\nu/2} \sum_{k=0}^{\infty} \frac{\left(\frac{\nu}{2}\right)_k \left(\frac{\nu+\alpha_1}{2}+1\right)_k \left(\frac{\nu+\alpha_2}{2}+1\right)_k}{k!\left(\frac{\nu+\alpha_1}{2}\right)_k \left(\frac{\nu+\alpha_2}{2}\right)_k}\rho^{2k}\right.$$

$$\left. \times \left(1 - \rho^2 \frac{\nu + 2k}{\nu + \alpha_1 + 2k}\right)\left(1 - \rho^2 \frac{\nu + 2k}{\nu + \alpha_2 + 2k}\right) - 1\right].$$

The proofs of parts (a) and (b) are trivial. For parts (c) and (d), the Euler formula (9.131.1.11) of [23] is used, $F_{2,1}(a, b, c; x) = (1 - x)^{c-b-a} F_{2,1}(c - a, c - b, c; x)$, and Proposition 2 with $a = b = 1$.

Figure 2 shows the correlation $\rho_{\mathbf{Y}}$ of Corollary 1d for some pdf parameters of $\mathbf{Y}$ (6). For all cases, when $\alpha_1$ and $\alpha_2$ increase, the correlation $\rho_{\mathbf{Y}}$ decreases. When $\nu$ increases, correlation $\rho_{\mathbf{Y}}$ slowly decreases from small to large values of $\alpha_1$ and $\alpha_2$. When parameter $\rho$ (the normal distribution correlation) increases, the correlation $\rho_{\mathbf{Y}}$ increases, as does its maximum value (from 0.06 to 0.55). We can observe in Corollary 1d that correlation $\rho_{\mathbf{Y}}$ does not depend on $\beta$.

The following Proposition 3 is useful for computing differential entropy and mutual information index of Section 3.

**Proposition 3.** *If* $\mathbf{Y} = (Y_1, Y_2)^\top$ *have pdf given in* (6), *then*

*(a)*

$$\mathbb{E}_{\mathbf{Y}}[Y_i] = \frac{2(1-\rho^2)^{(\nu+\alpha_i)/2+1}\Gamma\left(\frac{\nu+\alpha_i}{2}+1\right)}{\beta\Gamma\left(\frac{\nu+\alpha_i}{2}\right)} \sum_{k=0}^{\infty} \frac{\left(\frac{\nu}{2}\right)_k\left(\frac{\nu+\alpha_i}{2}+1\right)_k\rho^{2k}}{k!\left(\frac{\nu+\alpha_i}{2}\right)_k} F_{2,1}\left(\frac{\alpha_i}{2}, \frac{\nu+\alpha_i}{2}+1+k; \frac{\nu+\alpha_i}{2}+k; \rho^2\right), \quad (11)$$

$$i, j = 1, 2, \ i \neq j.$$

*(b)*

$$\mathbb{E}_{\mathbf{Y}}[\log Y_i] = \begin{cases} (1-\rho^2)^{(\nu+\alpha_i)/2} \displaystyle\sum_{k=0}^{\infty}\sum_{m_1=0}^{\infty} \frac{\left(\frac{\nu}{2}\right)_k\left(\frac{\alpha_i}{2}\right)_{m_1}\rho^{2(k+m_1)}}{k!m_1!}\left[\psi\left(\frac{\nu+\alpha_i}{2}+k+m_1\right) - \log\left(\frac{\beta}{2(1-\rho^2)}\right)\right], & if \ \rho > 0; \\ \psi\left(\frac{\nu+\alpha_i}{2}\right) - \log\left(\frac{\beta}{2}\right), & if \ \rho = 0. \end{cases} \quad (12)$$

$$i, j = 1, 2, \ i \neq j; \ where \ \psi(x) = \frac{d}{dx}\log\Gamma(x) \ is \ the \ digamma \ function.$$

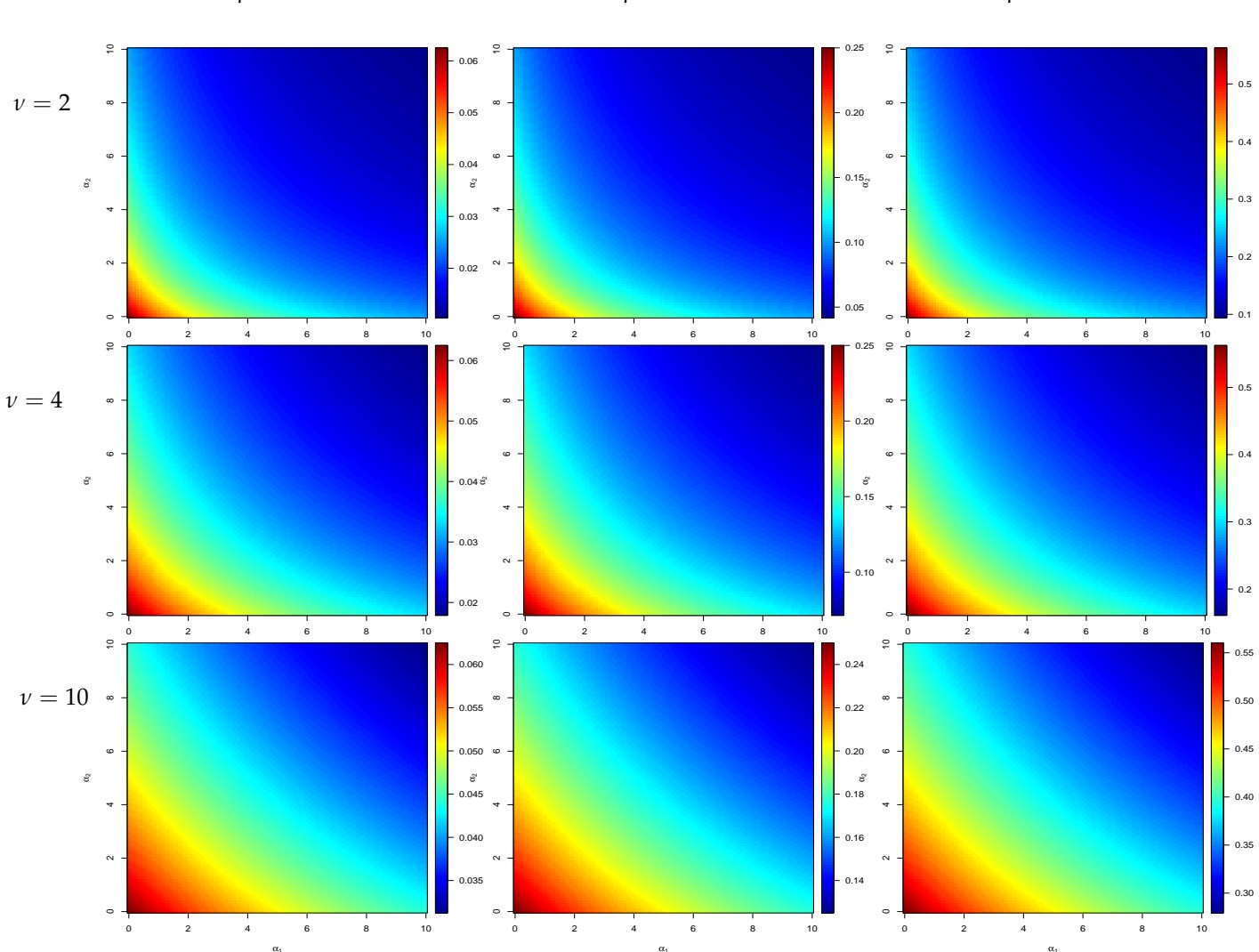

**Figure 2.** Correlation $\rho_{\mathbf{Y}}$ of Corollary 1d for some parameter combinations.

### 2.2. Hazard, Bonferroni and Lorenz Functions

The Bonferroni and Lorenz curves [24] have many practical applications not only in economy, but also in fields like reliability, lifetime testing, insurance, and medicine.

For random vector $\mathbf{Y} = (Y_1, Y_2)^\top$ with cdf $F(\mathbf{y}) = F_\mathbf{Y}(Y_1 \le y_1, Y_2 \le y_2)$, the Hazard, Bonferroni and Lorenz functions are defined by

$$\mathcal{Z}_\mathbf{Y}(\mathbf{t}) = \frac{f_\mathbf{Y}(\mathbf{t})}{1 - F(\mathbf{t})}, \tag{13}$$

with $\mathbf{t} = (t_1, t_2)^\top$;

$$\mathcal{B}_q(\mathbf{Y}) = \frac{1}{F(\mathbf{y})E[\mathbf{Y}]} \int_0^q \int_0^q \mathbf{y} f_\mathbf{Y}(\mathbf{y}) dy_1 dy_2, \tag{14}$$

with $q$ a real scalar, $0 < q < +\infty$; and

$$\mathcal{L}_q(\mathbf{Y}) = \mathcal{B}_q(\mathbf{y}) E[\mathbf{Y}], \tag{15}$$

respectively.

The Bonferroni curve for pdf (6) considered $F(\mathbf{y})$ and $E[\mathbf{Y}]$, obtained from Theorem 2 and Proposition 2 (by replacing $a = b = 1$), respectively. The double integral of the left side of (14) is obtained with the following Proposition.

**Proposition 4.** *Let* $\mathbf{Y} = (Y_1, Y_2)^\top$ *be a random vector with pdf given in (6) and q a real scalar, $0 < q < +\infty$, thus*

$$\int_0^q \int_0^q \mathbf{y} f_\mathbf{Y}(\mathbf{y}) dy_1 dy_2 = \frac{4(1-\rho^2)^{(\nu+\alpha_1+\alpha_2)/2+2} \Gamma\left(\frac{\nu+\alpha_1}{2}+1\right)\Gamma\left(\frac{\nu+\alpha_2}{2}+1\right)}{\beta^2 \Gamma\left(\frac{\nu+\alpha_1}{2}\right)\Gamma\left(\frac{\nu+\alpha_2}{2}\right)}$$

$$\times \sum_{k=0}^\infty \frac{\left(\frac{\nu}{2}\right)_k \left(\frac{\nu+\alpha_1}{2}+1\right)\left(\frac{\nu+\alpha_2}{2}+1\right)\rho^{2k}}{k!\left(\frac{\nu+\alpha_1}{2}\right)\left(\frac{\nu+\alpha_2}{2}\right)}$$

$$\times \gamma_{2,1}\left(\frac{\alpha_1}{2}, \left(\frac{\nu+\alpha_1}{2}+1+k, \frac{\beta q}{2(1-\rho^2)}\right); \frac{\nu+\alpha_1}{2}+k; \rho^2\right)$$

$$\times \gamma_{2,1}\left(\frac{\alpha_2}{2}, \left(\frac{\nu+\alpha_2}{2}+1+k, \frac{\beta q}{2(1-\rho^2)}\right); \frac{\nu+\alpha_2}{2}+k; \rho^2\right),$$

*where* $\gamma_{2,1}(a_1, (a_2, x); b_1; z)$ *is denoted in Theorem 2.*

Given that $F(\mathbf{y})$ and $E[\mathbf{Y}]$ depend on incomplete and complete gaussian hypergeometric functions, respectively, the Hazard, Bonferroni and Lorenz curves can be computed using these functions.

## 3. Differential Entropy and Mutual Information Index

The differential entropy of a random variable $Y$ is a variation measure of information uncertainty [25]. In particular, the differential entropy of $\mathbf{Y} = (Y_1, Y_2)^\top$ with pdf $f_\mathbf{Y}(\mathbf{y})$ is defined by

$$\mathcal{H}(\mathbf{Y}) = -\mathbb{E}_\mathbf{Y}[\log\{f_\mathbf{Y}(\mathbf{Y})\}] = -\int_0^\infty \int_0^\infty f_\mathbf{Y}(\mathbf{y}) \log f_\mathbf{Y}(\mathbf{y}) dy_1 dy_2, \tag{16}$$

and measures the contained information in $\mathbf{Y}$ based on its pdf's parameters. The following Remarks 1 and 2 will be used in the Proposition 5 to approximate the differential entropy of $\mathbf{Y}$.

**Remark 1** ([26])**.** *For a positive and fixed* $y_i$, *and fixed parameters* $\alpha_k$, $k = 1, 2$, $\rho$ *and* $\beta$, *we have*

$$F_{1,1}\left(\frac{\alpha_i}{2}; \frac{\nu+\alpha_i}{2}; \frac{\beta\rho^2 y_i}{2(1-\rho^2)}\right) = \sum_{s=0}^{n-1} \frac{\left(\frac{\alpha_i}{2}\right)_s}{\left(\frac{\nu+\alpha_i}{2}\right)_s s!}\left(\frac{\beta\rho^2 y_2}{2(1-\rho^2)}\right)^s + \mathcal{O}(|\nu|^{-n}), \quad n = 1, 2, \ldots,$$

*as $|\nu| \to \infty$.*

**Remark 2** (Formula 1.511 of [23])**.** *If $n \to \infty$ and $x = u/n$ (i.e., $x$ turn around 0, $-1 < x \le 1$, and there exists a constant $\mu > 0$ such that $|\log(1 + u/n) - u/n| \le \mu/n^2$), see Lemma 3.2 of [27]), we get*

$$\log(1 + x) = x + \mathcal{O}(n^{-2}).$$

**Proposition 5.** *The differential entropy of a random vector $\mathbf{Y} = (Y_1, Y_2)^\top$ with probability density function given in (6) can be approximated as*

$$
\mathcal{H}(\mathbf{Y}) \approx -\log\left\{ \frac{\left(\frac{\beta}{2}\right)^{\nu + \frac{\alpha_1 + \alpha_2}{2}} (1 - \rho^2)^{-\nu/2}}{\Gamma\left(\frac{\nu + \alpha_1}{2}\right)\Gamma\left(\frac{\nu + \alpha_2}{2}\right)} \right\} - \left(\frac{\nu + \alpha_1}{2} - 1\right)\mathbb{E}_{\mathbf{Y}}[\log Y_1] - \left(\frac{\nu + \alpha_2}{2} - 1\right)\mathbb{E}_{\mathbf{Y}}[\log Y_2]
$$

$$
- \frac{\beta}{2(1 - \rho^2)}\left\{ \left(\rho^2 \frac{\left(\frac{\alpha_1}{2}\right)_1}{\left(\frac{\nu + \alpha_1}{2}\right)_1} - 1\right)\mathbb{E}_{\mathbf{Y}}[Y_1] + \left(\rho^2 \frac{\left(\frac{\alpha_2}{2}\right)_1}{\left(\frac{\nu + \alpha_2}{2}\right)_1} - 1\right)\mathbb{E}_{\mathbf{Y}}[Y_2] \right\},
$$

*where $\mathbb{E}_{\mathbf{Y}}[Y_k]$ and $\mathbb{E}_{\mathbf{Y}}[\log Y_k]$, $k = 1, 2$, can be computed using parts (a) and (b) of Proposition 3, respectively.*

Under dependence assumption ($\rho \ne 0$), the mutual information index (MII) [25,28,29] between $Y_1$ and $Y_2$ is defined by

$$
\mathcal{M}(Y_1, Y_2) = E\left[\log\left\{ \frac{f_{\mathbf{Y}}(y_1, y_2)}{f_{Y_1}(y_1)f_{Y_2}(y_2)} \right\}\right] = \int_0^\infty \int_0^\infty f_{\mathbf{Y}}(y_1, y_2)\log\left\{ \frac{f_{\mathbf{Y}}(y_1, y_2)}{f_{Y_1}(y_1)f_{Y_2}(y_2)} \right\}dy_1 dy_2. \tag{17}
$$

It is clear from (17) that MII between $Y_1$ and $Y_2$ can be expressed in terms of marginal and joint differential entropies, $\mathcal{M}(\mathbf{Y}) = \mathcal{H}(Y_1) + \mathcal{H}(Y_2) - \mathcal{H}(\mathbf{Y})$ [25]. According to (5), the differential entropy of each gamma distribution is

$$
\mathcal{H}(Y_k) = \frac{\alpha_k + \nu}{2} - \log\left(\frac{\beta}{2}\right) + \log\Gamma\left(\frac{\alpha_k + \nu}{2}\right) + \left(1 - \frac{\alpha_k + \nu}{2}\right)\psi\left(\frac{\alpha_k + \nu}{2}\right), \quad k = 1, 2. \tag{18}
$$

Therefore, the MII between $Y_1$ and $Y_2$ can be computed using (18) and Proposition (5). Under independence assumption $\rho = 0$, the mutual information index between $Y_1$ and $Y_2$ is 0; otherwise, this index is positive [28,29]. Moreover, the MII increases with the degree of dependence between the components of $Y_1$ and $Y_2$. Therefore, the MII is an association measure between $Y_1$ and $Y_2$, which could be compared with correlation $\rho_{\mathbf{Y}}$.

Figure 3 illustrates the behavior of MII assuming several values for parameters of $\mathbf{Y}$. The case $\rho = 0$ was omitted for the above mentioned reasons, and the case $\rho = 0.5$ was omitted because results are similar to the case $\rho = 0.25$. We observed that $\mathcal{M}(Y_1, Y_2) < 0$ for small values of $\beta$, which is related to approximation (A16) being wrongly utilized when the argument is outside $(-1, 1]$ in Remark 2. However, when the argument is inside $(-1, 1]$, we get $\mathcal{M}(Y_1, Y_2) > 0$ and increases for large values of $\beta$ and any values of $\nu$, as in the analysis of $\rho_{\mathbf{Y}}$ in Figure 2.

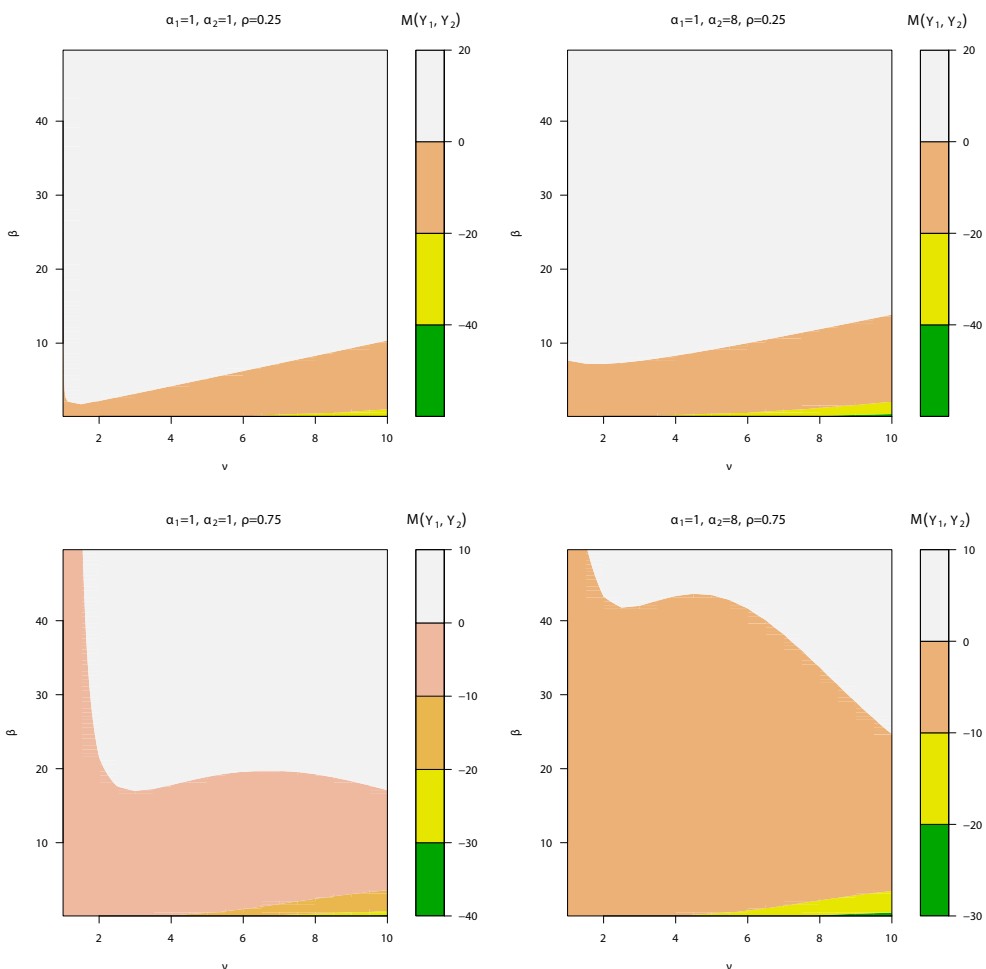

**Figure 3.** Mutual information index for several parameters of density given in (6).

## 4. Concluding Remarks

In this paper, we presented a generalization of bivariate gamma distribution based on a Kibble type bivariate gamma distribution. The stochastic representation was obtained by the sum of a Kibble-type bivariate random vector and a bivariate random vector builded by two independent gamma random variables. Moreover, the resulting bivariate density considers an infinite series of products of two confluent hypergeometric functions. In particular, we derived the probability and cumulative distribution functions, moment generation and characteristic functions, covariance, correlation and cross-product moment, Hazard, Bonferroni and Lorenz functions, and an approximation for the differential entropy and mutual information index. Numerical examples showed the behavior of exact and approximated expressions.

Previous work by [30] considered the generalization of this paper to represent bivariate Superstatistics based on Boltzmann factors. However, further work derived from this study could extend to the multivariate case (*d*-dimensional). A possible extension is considering Equation (6) of [31] for the joint pdf of our vector $(W_1, \ldots, W_d)^\top$, corresponding to a pdf based on a gamma distribution with simple Markov chain-type correlation. When $d = 2$, this pdf coincides with Kibble distribution defined in (4). Thus,

$$Y_1 = W_1 + R_1$$

$$\vdots$$

$$Y_d = W_d + R_d,$$

where $R_1, \ldots, R_d, R_i \sim Gamma(\alpha_i/2, \beta/2)$, $i = 1, \ldots, d$ are independent and gamma distributed random variables. However, the properties obtained in this study could be difficult to obtain for this multivariate version and the use of generalized hypergeometric function must be carefully handled. In addition, it is possible to consider the Probability Transformation method using the theory of the space transformation of random variables and the probability conservation principle. Thus, we could evaluate the pdf of a $d$-dimensional invertible transformation [32].

Inferential aspects could also be considered in future work. For example: (i) a numerical approach could be used in the optimization of log-likelihood function; (ii) the pseudo-likelihood method by considering the optimization of an objective function that depends on a bivariate pdf could be used; and (iii) a Bayesian approach could be useful.

**Author Contributions:** C.C.-C. and J.E.C.-R. wrote the paper and contributed reagents/analysis/materials tools; C.C.-C. and J.E.C.-R. conceived, designed and performed the experiments. All authors have read and approved the final manuscript.

**Funding:** Caamaño-Carrillo's research was funded by FONDECYT (Chile) grant No. 11220066 and by Proyecto Regular DIUBB 2120538 IF/R de la Universidad del Bío-Bío. Contreras-Reyes's research was funded by FONDECYT (Chile) grant No. 11190116.

**Institutional Review Board Statement:** Not applicable.

**Informed Consent Statement:** Not applicable.

**Data Availability Statement:** Not applicable.

**Acknowledgments:** The authors thank the editor and three anonymous referees for their helpful comments and suggestions.

**Conflicts of Interest:** The authors declare that there is no conflict of interest in the publication of this paper.

## Appendix A

**Proof of Theorem 1.** Let $w_1 = y_1 - r_1$ and $w_2 = y_2 - r_2$ be a transformation in (4), $0 < r_k < y_k$, $k = 1, 2$, with Jacobian $J((w_1, w_2) \to (y_1, y_2)) = 1$. Using series expansion of the hypergeometric function $F_{0,1}$, we get

$$
\begin{aligned}
f_{\mathbf{Y}}(\mathbf{y}) &= \int_0^{y_1} \int_0^{y_2} f_{\mathbf{W}|\mathbf{R}}(\mathbf{w}|\mathbf{r}) f_{\mathbf{R}}(\mathbf{r}) |J| d\mathbf{r} \\
&= \int_0^{y_1} \int_0^{y_2} \frac{\left(\frac{\beta}{2}\right)^{\nu + \frac{\alpha_1 + \alpha_2}{2}} [(y_1 - r_1)(y_2 - r_2)]^{\nu/2 - 1} r_1^{\alpha_1/2 - 1} r_2^{\alpha_2/2 - 1} e^{-\frac{\beta}{2(1-\rho^2)}[(y_1 - r_1) + (y_2 - r_2)]} e^{-\frac{\beta}{2}(r_1 + r_2)}}{\Gamma^2\left(\frac{\nu}{2}\right) \Gamma\left(\frac{\alpha_1}{2}\right) \Gamma\left(\frac{\alpha_2}{2}\right) (1 - \rho^2)^{\nu/2}} \\
&\quad \times F_{0,1}\left(\frac{\nu}{2}; \frac{\beta^2 \rho^2 (y_1 - r_1)(y_2 - r_2)}{4(1-\rho^2)^2}\right) d\mathbf{r} \\
&= \sum_{k=0}^{\infty} \int_0^{y_1} \int_0^{y_2} \frac{\left(\frac{\beta}{2}\right)^{\nu + \frac{\alpha_1 + \alpha_2}{2}} [(y_1 - r_1)(y_2 - r_2)]^{\nu/2 + k - 1} r_1^{\alpha_1/2 - 1} r_2^{\alpha_2/2 - 1} e^{-\frac{\beta}{2(1-\rho^2)}(y_1 + y_2)} e^{\frac{\beta \rho^2}{2(1-\rho^2)}(r_1 + r_2)}}{\Gamma^2\left(\frac{\nu}{2}\right) \Gamma^2\left(\frac{\alpha}{2}\right) (1 - \rho^2)^{\nu/2}} \\
&\quad \times \frac{1}{k! \left(\frac{\nu}{2}\right)_k} \left(\frac{\beta^2 \rho^2}{4(1-\rho^2)^2}\right)^k d\mathbf{r} \\
&= \frac{\left(\frac{\beta}{2}\right)^{\nu + \frac{\alpha_1 + \alpha_2}{2}} e^{-\frac{\beta}{2(1-\rho^2)}(y_1 + y_2)}}{\Gamma^2\left(\frac{\nu}{2}\right) \Gamma\left(\frac{\alpha_1}{2}\right) \Gamma\left(\frac{\alpha_2}{2}\right) (1 - \rho^2)^{\nu/2}} \sum_{k=0}^{\infty} \frac{I(k)}{k! \left(\frac{\nu}{2}\right)_k} \left(\frac{\beta^2 \rho^2}{4(1-\rho^2)^2}\right)^k.
\end{aligned}
\tag{A1}
$$

Then, using Fubini's Theorem and formula (3.383.1.11) of [23], we obtain

$$I(k) = \int\limits_0^{y_1} (y_1 - r_1)^{\nu/2+k-1} r_1^{\alpha_1/2-1} e^{\frac{\beta\rho^2}{2(1-\rho^2)} r_1} dr_1 \int\limits_0^{y_2} (y_2 - r_2)^{\nu/2+k-1} r_2^{\alpha_2/2-1} e^{\frac{\beta\rho^2}{2(1-\rho^2)} r_2} dr_2$$

$$= \frac{\Gamma^2\left(\frac{\nu}{2}+k\right)\Gamma\left(\frac{\alpha_1}{2}\right)\Gamma\left(\frac{\alpha_2}{2}\right)}{\Gamma\left(\frac{\nu+\alpha_1}{2}+k\right)\Gamma\left(\frac{\nu+\alpha_2}{2}+k\right)} y_1^{\frac{(\nu+\alpha_1)}{2}+k-1} y_2^{\frac{(\nu+\alpha_2)}{2}+k-1} F_{1,1}\left(\frac{\alpha_1}{2}; \frac{\nu+\alpha_1}{2}+k; \frac{\beta\rho^2 y_1}{2(1-\rho^2)}\right) \tag{A2}$$

$$\times F_{1,1}\left(\frac{\alpha_2}{2}; \frac{\nu+\alpha_2}{2}+k; \frac{\beta\rho^2 y_2}{2(1-\rho^2)}\right).$$

Combining Equations (A2) and (A1), the result is obtained. □

**Proof of Theorem 2.** Using series expansion of the hypergeometric function $F_{1,1}$, we get

$$F_{\mathbf{Y}}(Y_1 \le t_1, Y_2 \le t_2) = \int\limits_0^{t_1} \int\limits_0^{t_2} f_{\mathbf{Y}}(\mathbf{y}) d\mathbf{y}$$

$$= \frac{\left(\frac{\beta}{2}\right)^{\nu+\frac{\alpha_1+\alpha_2}{2}}}{\Gamma\left(\frac{\nu+\alpha_1}{2}\right)\Gamma\left(\frac{\nu+\alpha_2}{2}\right)(1-\rho^2)^{\nu/2}} \int\limits_0^{t_1}\int\limits_0^{t_2} y_1^{(\nu+\alpha_1)/2-1} y_2^{(\nu+\alpha_2)/2-1} e^{-\frac{\beta}{2(1-\rho^2)}(y_1+y_2)}$$

$$\times \sum_{k=0}^\infty \frac{\left(\frac{\nu}{2}\right)_k}{k!\left(\frac{\nu+\alpha_1}{2}\right)_k \left(\frac{\nu+\alpha_2}{2}\right)_k} \left(\frac{\beta^2\rho^2 y_1 y_2}{4(1-\rho^2)^2}\right)^k F_{1,1}\left(\frac{\alpha_1}{2}; \frac{\nu+\alpha_1}{2}+k; \frac{\beta\rho^2 y_1}{2(1-\rho^2)}\right)$$

$$\times F_{1,1}\left(\frac{\alpha_2}{2}; \frac{\nu+\alpha_2}{2}+k; \frac{\beta\rho^2 y_2}{2(1-\rho^2)}\right) d\mathbf{y} \tag{A3}$$

$$= \frac{\left(\frac{\beta}{2}\right)^{\nu+\frac{\alpha_1+\alpha_2}{2}}}{\Gamma\left(\frac{\nu+\alpha_1}{2}\right)\Gamma\left(\frac{\nu+\alpha_2}{2}\right)(1-\rho^2)^{\nu/2}} \sum_{k=0}^\infty \sum_{m_1=0}^\infty \sum_{m_2=0}^\infty \frac{\left(\frac{\nu}{2}\right)_k I(k,m_1,m_2)}{k!\left(\frac{\nu+\alpha_1}{2}\right)_k \left(\frac{\nu+\alpha_2}{2}\right)_k} \left(\frac{\beta^2\rho^2}{4(1-\rho^2)^2}\right)^k$$

$$\times \frac{\left(\frac{\alpha_1}{2}\right)_{m_1}}{m_1!\left(\frac{\nu+\alpha_1}{2}+k\right)_{m_1}} \left(\frac{\beta\rho^2}{2(1-\rho^2)}\right)^{m_1} \frac{\left(\frac{\alpha_2}{2}\right)_{m_2}}{m_2!\left(\frac{\nu+\alpha_2}{2}+k\right)_{m_2}} \left(\frac{\beta\rho^2}{2(1-\rho^2)}\right)^{m_2}.$$

Using Fubini's Theorem and formula (3.381.1) of [23], we obtain

$$I(k,m_1,m_2) = \int\limits_0^{t_1} y_1^{(\nu+\alpha_1)/2+k+m_1-1} e^{-\frac{\beta y_1}{2(1-\rho^2)}} dy_1 \int\limits_0^{t_1} y_2^{(\nu+\alpha_2)/2+k+m_2-1} e^{-\frac{\beta y_2}{2(1-\rho^2)}} dy_2$$

$$= \left(\frac{2(1-\rho^2)}{\beta}\right)^{(\nu+\alpha_1)/2+k+m_1} \left(\frac{2(1-\rho^2)}{\beta}\right)^{(\nu+\alpha_2)/2+k+m_2} \tag{A4}$$

$$\times \gamma\left(\frac{\nu+\alpha_1}{2}+k+m_1, \frac{\beta t_1}{2(1-\rho^2)}\right) \gamma\left(\frac{\nu+\alpha_2}{2}+k+m_2, \frac{\beta t_2}{2(1-\rho^2)}\right),$$

where $\gamma(a,x) = \int_{-\infty}^x e^{-t} t^{a-1} dt$, $a > 0$ is the lower incomplete gamma function. Combining Equations (A3) and (A4), we obtain

$$F_{\mathbf{Y}}(Y_1 \le t_1, Y_2 \le t_2) = \frac{(1-\rho^2)^{(\nu+\alpha_1+\alpha_2)/2}}{\Gamma\left(\frac{\nu+\alpha_1}{2}\right)\Gamma\left(\frac{\nu+\alpha_2}{2}\right)} \sum_{k=0}^{\infty} \frac{\left(\frac{\nu}{2}\right)_k \rho^{2k}}{k!\left(\frac{\nu+\alpha_1}{2}\right)_k\left(\frac{\nu+\alpha_2}{2}\right)_k}$$

$$\times \sum_{m_1=0}^{\infty} \frac{\left(\frac{\alpha_1}{2}\right)_{m_1} \gamma\left(\frac{\nu+\alpha_1}{2}+k+m_1, \frac{\beta t_1}{2(1-\rho^2)}\right)\rho^{2m_1}}{m_1!\left(\frac{\nu+\alpha_1}{2}+k\right)_{m_1}} \sum_{m_2=0}^{\infty} \frac{\left(\frac{\alpha_2}{2}\right)_{m_2} \gamma\left(\frac{\nu+\alpha_2}{2}+k+m_2, \frac{\beta t_2}{2(1-\rho^2)}\right)\rho^{2m_2}}{m_2!\left(\frac{\nu+\alpha_2}{2}+k\right)_{m_2}}$$

$$= \frac{(1-\rho^2)^{(\nu+\alpha_1+\alpha_2)/2}}{\Gamma\left(\frac{\nu+\alpha_1}{2}\right)\Gamma\left(\frac{\nu+\alpha_2}{2}\right)} \sum_{k=0}^{\infty} \frac{\left(\frac{\nu}{2}\right)_k \rho^{2k}}{k!\left(\frac{\nu+\alpha_1}{2}\right)_k\left(\frac{\nu+\alpha_2}{2}\right)_k}$$

$$\times \sum_{m_1=0}^{\infty} \frac{\left(\frac{\alpha_1}{2}\right)_{m_1} \left(\frac{\nu+\alpha_1}{2}+k; \frac{\beta t_1}{2(1-\rho^2)}\right)_{m_1} \Gamma\left(\frac{\nu+\alpha_1}{2}+k\right)\rho^{2m_1}}{m_1!\left(\frac{\nu+\alpha_1}{2}+k\right)_{m_1}} \times \sum_{m_2=0}^{\infty} \frac{\left(\frac{\alpha_2}{2}\right)_{m_2} \left(\frac{\nu+\alpha_2}{2}+k; \frac{\beta t_2}{2(1-\rho^2)}\right)_{m_2} \Gamma\left(\frac{\nu+\alpha_2}{2}+k\right)\rho^{2m_2}}{m_2!\left(\frac{\nu+\alpha_2}{2}+k\right)_{m_2}}$$

$$= (1-\rho^2)^{(\nu+\alpha_1+\alpha_2)/2} \sum_{k=0}^{\infty} \frac{\left(\frac{\nu}{2}\right)_k \rho^{2k}}{k!} \gamma_{2,1}\left(\frac{\alpha_1}{2}, \left(\frac{\nu+\alpha_1}{2}+k, \frac{\beta t_1}{2(1-\rho^2)}\right); \frac{\nu+\alpha_1}{2}+k; \rho^2\right)$$

$$\times \gamma_{2,1}\left(\frac{\alpha_2}{2}, \left(\frac{\nu+\alpha_2}{2}+k, \frac{\beta t_2}{2(1-\rho^2)}\right); \frac{\nu+\alpha_2}{2}+k; \rho^2\right).$$

This concludes the proof. □

**Proof of Proposition 1.** By definition of mfg and using series expansion of hypergeometric function $F_{1,1}$, we get

$$M(t_1, t_2) = \frac{\left(\frac{\beta}{2}\right)^{\nu+\frac{\alpha_1+\alpha_2}{2}}}{\Gamma\left(\frac{\nu+\alpha_1}{2}\right)\Gamma\left(\frac{\nu+\alpha_2}{2}\right)(1-\rho^2)^{\nu/2}} \sum_{k=0}^{\infty}\sum_{m_1=0}^{\infty}\sum_{m_2=0}^{\infty} \frac{\left(\frac{\nu}{2}\right)_k I(k, m_1, m_2)}{k!\left(\frac{\nu+\alpha_1}{2}\right)_k\left(\frac{\nu+\alpha_2}{2}\right)_k} \left(\frac{\beta^2\rho^2}{4(1-\rho^2)^2}\right)^k$$

$$\times \frac{\left(\frac{\alpha_1}{2}\right)_{m_1}}{m_1!\left(\frac{\nu+\alpha_1}{2}+k\right)_{m_1}} \left(\frac{\beta\rho^2}{2(1-\rho^2)}\right)^{m_1} \frac{\left(\frac{\alpha_2}{2}\right)_{m_2}}{m_2!\left(\frac{\nu+\alpha_2}{2}+k\right)_{m_2}} \left(\frac{\beta\rho^2}{2(1-\rho^2)}\right)^{m_2}. \tag{A5}$$

Using Fubini's Theorem and formula (3.381.4) of [23], we obtain

$$I(k, m_1, m_2) = \int_0^{\infty} y_1^{(\nu+\alpha_1)/2+k+m_1-1} e^{-\frac{\beta-2(1-\rho^2)t_1}{2(1-\rho^2)}y_1} dy_1 \int_0^{\infty} y_2^{(\nu+\alpha_2)/2+k+m_2-1} e^{-\frac{\beta-2(1-\rho^2)t_2}{2(1-\rho^2)}y_2} dy_2$$

$$= \left(\frac{2(1-\rho^2)}{\beta-2(1-\rho^2)t_1}\right)^{(\nu+\alpha_1)/2+k+m_1} \left(\frac{2(1-\rho^2)}{\beta-2(1-\rho^2)t_2}\right)^{(\nu+\alpha_2)/2+k+m_2}$$

$$\times \Gamma\left(\frac{\nu+\alpha_1}{2}+k+m_1\right)\Gamma\left(\frac{\nu+\alpha_2}{2}+k+m_2\right). \tag{A6}$$

Combining Equations (A5) and (A6), we obtain

$$M(t_1, t_2) = \frac{\beta^{\nu+(\alpha_1+\alpha_2)/2}(1-\rho^2)^{(\nu+\alpha_1+\alpha_2)/2}}{\Gamma\left(\frac{\nu+\alpha_1}{2}\right)\Gamma\left(\frac{\nu+\alpha_2}{2}\right)[\beta-2(1-\rho^2)t_1]^{(\nu+\alpha_1)/2}[\beta-2(1-\rho^2)t_2]^{(\nu+\alpha_2)/2}} \sum_{k=0}^{\infty} \frac{\left(\frac{\nu}{2}\right)_k}{k!\left(\frac{\nu+\alpha_1}{2}\right)_k\left(\frac{\nu+\alpha_2}{2}\right)_k} \left(\frac{\beta^2\rho^2}{\beta-2(1-\rho^2)t_1}\right)^k$$

$$\times \sum_{m_1=0}^{\infty} \frac{\left(\frac{\alpha_1}{2}\right)_{m_1}\Gamma\left(\frac{\nu+\alpha_1}{2}+k+m_1\right)}{m_1!\left(\frac{\nu+\alpha_1}{2}+k\right)_{m_1}} \left(\frac{\beta\rho^2}{\beta-2(1-\rho^2)t_1}\right)^{m_1} \sum_{m_2=0}^{\infty} \frac{\left(\frac{\alpha_2}{2}\right)_{m_2}\Gamma\left(\frac{\nu+\alpha_2}{2}+k+m_2\right)}{m_2!\left(\frac{\nu+\alpha_2}{2}+k\right)_{m_2}} \left(\frac{\beta\rho^2}{\beta-2(1-\rho^2)t_2}\right)^{m_2}$$

$$= \frac{\beta^{\nu+(\alpha_1+\alpha_2)/2}(1-\rho^2)^{(\nu+\alpha_1+\alpha_2)/2}}{[\beta-2(1-\rho^2)t_1]^{(\nu+\alpha_1)/2}[\beta-2(1-\rho^2)t_2]^{(\nu+\alpha_2)/2}} \sum_{k=0}^{\infty} \frac{\left(\frac{\nu}{2}\right)_k}{k!} \left(\frac{\beta^2\rho^2}{[\beta-2(1-\rho^2)t_1][\beta-2(1-\rho^2)t_2]}\right)^k$$

$$\times \sum_{m_1=0}^{\infty} \frac{\left(\frac{\alpha_1}{2}\right)_{m_1}}{m_1!} \left(\frac{\beta\rho^2}{\beta-2(1-\rho^2)t_1}\right)^{m_1} \sum_{m_2=0}^{\infty} \frac{\left(\frac{\alpha_2}{2}\right)_{m_2}}{m_2!} \left(\frac{\beta\rho^2}{\beta-2(1-\rho^2)t_2}\right)^{m_2}.$$

Considering $\sum_{k=0}^{\infty} \frac{(a)_k x^k}{k!} = (1-x)^{-a}$, the last equality yields

$$M(t_1, t_2) = \frac{\beta^{\nu+(\alpha_1+\alpha_2)/2}(1-\rho^2)^{(\nu+\alpha_1+\alpha_2)/2}}{[\beta - 2(1-\rho^2)t_1]^{(\nu+\alpha_1)/2}[\beta - 2(1-\rho^2)t_2]^{(\nu+\alpha_2)/2}} \left(1 - \frac{\beta^2\rho^2}{[\beta - 2(1-\rho^2)t_1][\beta - 2(1-\rho^2)t_2]}\right)^{-\frac{\nu}{2}}$$

$$\times \left(1 - \frac{\beta\rho^2}{\beta - 2(1-\rho^2)t_1}\right)^{-\frac{\alpha_1}{2}} \left(1 - \frac{\beta\rho^2}{\beta - 2(1-\rho^2)t_2}\right)^{-\frac{\alpha_2}{2}}.$$

This concludes the proof. $\square$

**Proof of Proposition 2.** By definition of cross-product moment and using series expansion of the hypergeometric function $F_{1,1}$, we get

$$\mathbb{E}(Y_1^a Y_2^b) = \frac{\left(\frac{\beta}{2}\right)^{\nu+\frac{\alpha_1+\alpha_2}{2}}}{\Gamma\left(\frac{\nu+\alpha_1}{2}\right)\Gamma\left(\frac{\nu+\alpha_2}{2}\right)(1-\rho^2)^{\nu/2}} \int_0^{\infty}\int_0^{\infty} y_1^{(\nu+\alpha_1)/2+a-1} y_2^{(\nu+\alpha_2)/2+b-1} e^{-\frac{\beta}{2(1-\rho^2)}(y_1+y_2)}$$

$$\times \sum_{k=0}^{\infty} \frac{\left(\frac{\nu}{2}\right)_k}{k!\left(\frac{\nu+\alpha_1}{2}\right)_k\left(\frac{\nu+\alpha_2}{2}\right)_k} \left(\frac{\beta^2\rho^2 y_1 y_2}{4(1-\rho^2)^2}\right)^k F_{1,1}\left(\frac{\alpha_1}{2}; \frac{\nu+\alpha_1}{2}+k; \frac{\beta\rho^2 y_1}{2(1-\rho^2)}\right) F_{1,1}\left(\frac{\alpha_2}{2}; \frac{\nu+\alpha_2}{2}+k; \frac{\beta\rho^2 y_2}{2(1-\rho^2)}\right) d\mathbf{y}$$

$$= \frac{\left(\frac{\beta}{2}\right)^{\nu+\frac{\alpha_1+\alpha_2}{2}}}{\Gamma\left(\frac{\nu+\alpha_1}{2}\right)\Gamma\left(\frac{\nu+\alpha_2}{2}\right)(1-\rho^2)^{\nu/2}} \sum_{k=0}^{\infty}\sum_{m_1=0}^{\infty}\sum_{m_2=0}^{\infty} \frac{\left(\frac{\nu}{2}\right)_k I(k,m_1,m_2)}{k!\left(\frac{\nu+\alpha_1}{2}\right)_k\left(\frac{\nu+\alpha_2}{2}\right)_k} \left(\frac{\beta^2\rho^2}{4(1-\rho^2)^2}\right)^k \tag{A7}$$

$$\times \frac{\left(\frac{\alpha_1}{2}\right)_{m_1}}{m_1!\left(\frac{\nu+\alpha_1}{2}+k\right)_{m_1}} \left(\frac{\beta\rho^2}{2(1-\rho^2)}\right)^{m_1} \frac{\left(\frac{\alpha_2}{2}\right)_{m_2}}{m_2!\left(\frac{\nu+\alpha_2}{2}+k\right)_{m_2}} \left(\frac{\beta\rho^2}{2(1-\rho^2)}\right)^{m_2}.$$

Using Fubini's Theorem and formula (3.381.4) of [23], we obtain

$$I(k,m_1,m_2) = \int_0^{\infty} y_1^{(\nu+\alpha_1)/2+a+k+m_1-1} e^{-\frac{\beta y_1}{2(1-\rho^2)}} dy_1 \int_0^{\infty} y_2^{(\nu+\alpha_2)/2+b+k+m_2-1} e^{-\frac{\beta y_2}{2(1-\rho^2)}} dy_2$$

$$= \left(\frac{2(1-\rho^2)}{\beta}\right)^{(\nu+\alpha_1)/2+a+k+m_1} \left(\frac{2(1-\rho^2)}{\beta}\right)^{(\nu+\alpha_2)/2+b+k+m_2} \Gamma\left(\frac{\nu+\alpha_1}{2}+a+k+m_1\right)\Gamma\left(\frac{\nu+\alpha_2}{2}+b+k+m_2\right). \tag{A8}$$

Combining Equations (A7) and (A8), we obtain

$$\mathbb{E}(Y_1^a Y_2^b) = \left(\frac{2}{\beta}\right)^{a+b} \frac{(1-\rho^2)^{(\nu+\alpha_1+\alpha_2)/2+a+b}}{\Gamma\left(\frac{\nu+\alpha_1}{2}\right)\Gamma\left(\frac{\nu+\alpha_2}{2}\right)} \sum_{k=0}^{\infty} \frac{\left(\frac{\nu}{2}\right)_k \rho^{2k}}{k!\left(\frac{\nu+\alpha_1}{2}\right)_k\left(\frac{\nu+\alpha_2}{2}\right)_k}$$

$$\times \sum_{m_1=0}^{\infty} \frac{\left(\frac{\alpha_1}{2}\right)_{m_1}\Gamma\left(\frac{\nu+\alpha_1}{2}+a+k+m_1\right)\rho^{2m_1}}{m_1!\left(\frac{\nu+\alpha_1}{2}+k\right)_{m_1}} \sum_{m_2=0}^{\infty} \frac{\left(\frac{\alpha_2}{2}\right)_{m_2}\Gamma\left(\frac{\nu+\alpha_2}{2}+b+k+m_2\right)\rho^{2m_2}}{m_2!\left(\frac{\nu+\alpha_2}{2}+k\right)_{m_2}}$$

$$= \left(\frac{2}{\beta}\right)^{a+b} \frac{(1-\rho^2)^{(\nu+\alpha_1+\alpha_2)/2+a+b}}{\Gamma\left(\frac{\nu+\alpha_1}{2}\right)\Gamma\left(\frac{\nu+\alpha_2}{2}\right)} \sum_{k=0}^{\infty} \frac{\left(\frac{\nu}{2}\right)_k \rho^{2k}}{k!\left(\frac{\nu+\alpha_1}{2}\right)_k\left(\frac{\nu+\alpha_2}{2}\right)_k}$$

$$\times \sum_{m_1=0}^{\infty} \frac{\left(\frac{\alpha_1}{2}\right)_{m_1}\left(\frac{\nu+\alpha_1}{2}+a+k\right)_{m_1}\Gamma\left(\frac{\nu+\alpha_1}{2}+a+k\right)\rho^{2m_1}}{m_1!\left(\frac{\nu+\alpha_1}{2}+k\right)_{m_1}} \sum_{m_2=0}^{\infty} \frac{\left(\frac{\alpha_2}{2}\right)_{m_2}\left(\frac{\nu+\alpha_2}{2}+k\right)_{m_2}\Gamma\left(\frac{\nu+\alpha_2}{2}+b+k\right)\rho^{2m_2}}{m_2!\left(\frac{\nu+\alpha_2}{2}+k\right)_{m_2}}.$$

This concludes the proof using basic algebra. $\square$

**Proof of Proposition 3.** For (a), we have

$$\mathbb{E}_{\mathbf{Y}}[Y_i] = \frac{\left(\frac{\beta}{2}\right)^{\nu+\frac{\alpha_i+\alpha_j}{2}}}{\Gamma\left(\frac{\nu+\alpha_i}{2}\right)\Gamma\left(\frac{\nu+\alpha_j}{2}\right)(1-\rho^2)^{\nu/2}} \sum_{k=0}^{\infty}\sum_{m_1=0}^{\infty}\sum_{m_2=0}^{\infty} \frac{\left(\frac{\nu}{2}\right)_k I(k,m_1,m_2)}{k!\left(\frac{\nu+\alpha_i}{2}\right)_k\left(\frac{\nu+\alpha_j}{2}\right)_k} \left(\frac{\beta^2\rho^2}{4(1-\rho^2)^2}\right)^k$$

$$\times \frac{\left(\frac{\alpha_i}{2}\right)_{m_1}}{m_1!\left(\frac{\nu+\alpha_i}{2}+k\right)_{m_1}} \left(\frac{\beta\rho^2}{2(1-\rho^2)}\right)^{m_1} \frac{\left(\frac{\alpha_j}{2}\right)_{m_2}}{m_2!\left(\frac{\nu+\alpha_j}{2}+k\right)_{m_2}} \left(\frac{\beta\rho^2}{2(1-\rho^2)}\right)^{m_2}. \tag{A9}$$

Using Fubini's Theorem and formula (3.381.4) of [23], we obtain

$$I(k,m_1,m_2) = \int_0^{\infty} y_i^{(\nu+\alpha_i)/2+k+m_1} e^{-\frac{\beta y_i}{2(1-\rho^2)}} dy_i \int_0^{\infty} y_j^{(\nu+\alpha_j)/2+k+m_1-1} e^{-\frac{\beta y_j}{2(1-\rho^2)}} dy_j$$

$$= \left(\frac{2(1-\rho^2)}{\beta}\right)^{\nu+(\alpha_i+\alpha_j)/2+2k+m_1+m_2+1} \Gamma\left(\frac{\nu+\alpha_i}{2}+k+m_1+1\right)\Gamma\left(\frac{\nu+\alpha_j}{2}+k+m_1\right). \tag{A10}$$

The proof is straightforward by combining Equations (A9) and (A10).

For (b), we have

$$\mathbb{E}_{\mathbf{Y}}[\log Y_i] = \frac{\left(\frac{\beta}{2}\right)^{\nu+\frac{\alpha_i+\alpha_j}{2}}}{\Gamma\left(\frac{\nu+\alpha_i}{2}\right)\Gamma\left(\frac{\nu+\alpha_j}{2}\right)(1-\rho^2)^{\nu/2}} \sum_{k=0}^{\infty}\sum_{m_1=0}^{\infty}\sum_{m_2=0}^{\infty} \frac{\left(\frac{\nu}{2}\right)_k I(k,m_1,m_2)}{k!\left(\frac{\nu+\alpha_i}{2}\right)_k\left(\frac{\nu+\alpha_j}{2}\right)_k} \left(\frac{\beta^2\rho^2}{4(1-\rho^2)^2}\right)^k$$

$$\times \frac{\left(\frac{\alpha_i}{2}\right)_{m_1}}{m_1!\left(\frac{\nu+\alpha_i}{2}+k\right)_{m_1}} \left(\frac{\beta\rho^2}{2(1-\rho^2)}\right)^{m_1} \frac{\left(\frac{\alpha_j}{2}\right)_{m_2}}{m_2!\left(\frac{\nu+\alpha_j}{2}+k\right)_{m_2}} \left(\frac{\beta\rho^2}{2(1-\rho^2)}\right)^{m_2}. \tag{A11}$$

Using Fubini's Theorem and formulas (3.381.4) and (4.352.1) of [23], we obtain

$$I(k,m_1,m2) = \int_0^{\infty} \log(y_i) y_i^{(\nu+\alpha_i)/2+k+m_1-1} e^{-\frac{\beta y_i}{2(1-\rho^2)}} dy_i \int_0^{\infty} y_j^{(\nu+\alpha_j)/2+k+m_2-1} e^{-\frac{\beta y_j}{2(1-\rho^2)}} dy_j$$

$$= \left(\frac{2(1-\rho^2)}{\beta}\right)^{\nu+(\alpha_i+\alpha_j)/2+2k+m_1+m_2} \Gamma\left(\frac{\nu+\alpha_i}{2}+k+m_1\right)\Gamma\left(\frac{\nu+\alpha_j}{2}+k+m_2\right)$$

$$\times \left[\psi\left(\frac{\nu+\alpha_i}{2}+k+m_1\right) - \log\left(\frac{\beta}{2(1-\rho^2)}\right)\right]. \tag{A12}$$

The proof is straightforward by combining Equations (A11) and (A12). □

**Proof of Proposition 4.** By replacing $a = b = 1$ in Proposition 2, we have

$$
\int_0^q \int_0^q \mathbf{y} f_{\mathbf{Y}}(\mathbf{y}) dy_1 dy_2 = \frac{\left(\frac{\beta}{2}\right)^{\nu+\frac{\alpha_1+\alpha_2}{2}}}{\Gamma\left(\frac{\nu+\alpha_1}{2}\right)\Gamma\left(\frac{\nu+\alpha_2}{2}\right)(1-\rho^2)^{\nu/2}} \int_0^q \int_0^q y_1^{(\nu+\alpha_1)/2} y_2^{(\nu+\alpha_2)/2} e^{-\frac{\beta}{2(1-\rho^2)}(y_1+y_2)}
$$

$$
\times \sum_{k=0}^{\infty} \frac{\left(\frac{\nu}{2}\right)_k}{k!\left(\frac{\nu+\alpha_1}{2}\right)_k \left(\frac{\nu+\alpha_2}{2}\right)_k} \left(\frac{\beta^2\rho^2 y_1 y_2}{4(1-\rho^2)^2}\right)^k F_{1,1}\left(\frac{\alpha_1}{2}; \frac{\nu+\alpha_1}{2}+k; \frac{\beta\rho^2 y_1}{2(1-\rho^2)}\right)
$$

$$
\times F_{1,1}\left(\frac{\alpha_2}{2}; \frac{\nu+\alpha_2}{2}+k; \frac{\beta\rho^2 y_2}{2(1-\rho^2)}\right) d\mathbf{y} \tag{A13}
$$

$$
= \frac{\left(\frac{\beta}{2}\right)^{\nu+\frac{\alpha_1+\alpha_2}{2}}}{\Gamma\left(\frac{\nu+\alpha_1}{2}\right)\Gamma\left(\frac{\nu+\alpha_2}{2}\right)(1-\rho^2)^{\nu/2}} \sum_{k=0}^{\infty} \sum_{m_1=0}^{\infty} \sum_{m_2=0}^{\infty} \frac{\left(\frac{\nu}{2}\right)_k I(k,m_1,m_2)}{k!\left(\frac{\nu+\alpha_1}{2}\right)_k \left(\frac{\nu+\alpha_2}{2}\right)_k} \left(\frac{\beta^2\rho^2}{4(1-\rho^2)^2}\right)^k
$$

$$
\times \frac{\left(\frac{\alpha_1}{2}\right)_{m_1}}{m_1!\left(\frac{\nu+\alpha_1}{2}+k\right)_{m_1}} \left(\frac{\beta\rho^2}{2(1-\rho^2)}\right)^{m_1} \frac{\left(\frac{\alpha_2}{2}\right)_{m_2}}{m_2!\left(\frac{\nu+\alpha_2}{2}+k\right)_{m_2}} \left(\frac{\beta\rho^2}{2(1-\rho^2)}\right)^{m_2}.
$$

Using Fubini's Theorem and formula (3.381.1) of [23], we obtain

$$
I(k,m_1,m_2) = \int_0^q y_1^{(\nu+\alpha_1)/2+k+m_1} e^{-\frac{\beta y_1}{2(1-\rho^2)}} dy_1 \int_0^q y_2^{(\nu+\alpha_2)/2+k+m_2} e^{-\frac{\beta y_2}{2(1-\rho^2)}} dy_2
$$

$$
= \left(\frac{2(1-\rho^2)}{\beta}\right)^{(\nu+\alpha_1)/2+1+k+m_1} \left(\frac{2(1-\rho^2)}{\beta}\right)^{(\nu+\alpha_2)/2+1+k+m_2} \tag{A14}
$$

$$
\times \gamma\left(\frac{\nu+\alpha_1}{2}+1+k+m_1, \frac{\beta q}{2(1-\rho^2)}\right) \gamma\left(\frac{\nu+\alpha_2}{2}+1+k+m_2, \frac{\beta q}{2(1-\rho^2)}\right),
$$

where $\gamma(\cdot,\cdot)$ is the lower incomplete gamma function. Combining Equations (A13) and (A14), we obtain

$$
\int_0^q \int_0^q \mathbf{y} f_{\mathbf{Y}}(\mathbf{y}) dy_1 dy_2 = \frac{4(1-\rho^2)^{(\nu+\alpha_1+\alpha_2)/2+2}}{\beta^2 \Gamma\left(\frac{\nu+\alpha_1}{2}\right)\Gamma\left(\frac{\nu+\alpha_2}{2}\right)} \sum_{k=0}^{\infty} \frac{\left(\frac{\nu}{2}\right)_k \rho^{2k}}{k!\left(\frac{\nu+\alpha_1}{2}\right)_k \left(\frac{\nu+\alpha_2}{2}\right)_k}
$$

$$
\times \sum_{m_1=0}^{\infty} \frac{\left(\frac{\alpha_1}{2}\right)_{m_1} \gamma\left(\frac{\nu+\alpha_1}{2}+1+k+m_1, \frac{\beta q}{2(1-\rho^2)}\right) \rho^{2m_1}}{m_1!\left(\frac{\nu+\alpha_1}{2}+k\right)_{m_1}} \sum_{m_2=0}^{\infty} \frac{\left(\frac{\alpha_2}{2}\right)_{m_2} \gamma\left(\frac{\nu+\alpha_2}{2}+1+k+m_2, \frac{\beta q}{2(1-\rho^2)}\right) \rho^{2m_2}}{m_2!\left(\frac{\nu+\alpha_2}{2}+k\right)_{m_2}}
$$

$$
= \frac{4(1-\rho^2)^{(\nu+\alpha_1+\alpha_2)/2+2}}{\beta^2 \Gamma\left(\frac{\nu+\alpha_1}{2}\right)\Gamma\left(\frac{\nu+\alpha_2}{2}\right)} \sum_{k=0}^{\infty} \frac{\left(\frac{\nu}{2}\right)_k \rho^{2k}}{k!\left(\frac{\nu+\alpha_1}{2}\right)_k \left(\frac{\nu+\alpha_2}{2}\right)_k}
$$

$$
\times \sum_{m_1=0}^{\infty} \frac{\left(\frac{\alpha_1}{2}\right)_{m_1} \left(\frac{\nu+\alpha_1}{2}+1+k; \frac{\beta q}{2(1-\rho^2)}\right)_{m_1} \Gamma\left(\frac{\nu+\alpha_1}{2}+1+k\right) \rho^{2m_1}}{m_1!\left(\frac{\nu+\alpha_1}{2}+k\right)_{m_1}}
$$

$$
\times \sum_{m_2=0}^{\infty} \frac{\left(\frac{\alpha_2}{2}\right)_{m_2} \left(\frac{\nu+\alpha_2}{2}+1+k; \frac{\beta q}{2(1-\rho^2)}\right)_{m_2} \Gamma\left(\frac{\nu+\alpha_2}{2}+1+k\right) \rho^{2m_2}}{m_2!\left(\frac{\nu+\alpha_2}{2}+k\right)_{m_2}}
$$

$$
= \frac{4(1-\rho^2)^{(\nu+\alpha_1+\alpha_2)/2+2} \Gamma\left(\frac{\nu+\alpha_1}{2}+1\right)\Gamma\left(\frac{\nu+\alpha_2}{2}+1\right)}{\beta^2 \Gamma\left(\frac{\nu+\alpha_1}{2}\right)\Gamma\left(\frac{\nu+\alpha_2}{2}\right)} \sum_{k=0}^{\infty} \frac{\left(\frac{\nu}{2}\right)_k \left(\frac{\nu+\alpha_1}{2}+1\right)\left(\frac{\nu+\alpha_2}{2}+1\right) \rho^{2k}}{k!\left(\frac{\nu+\alpha_1}{2}\right)\left(\frac{\nu+\alpha_2}{2}\right)}
$$

$$
\times \gamma_{2,1}\left(\frac{\alpha_1}{2}, \left(\frac{\nu+\alpha_1}{2}+1+k, \frac{\beta q}{2(1-\rho^2)}\right); \frac{\nu+\alpha_1}{2}+k; \rho^2\right) \gamma_{2,1}\left(\frac{\alpha_2}{2}, \left(\frac{\nu+\alpha_2}{2}+1+k, \frac{\beta q}{2(1-\rho^2)}\right); \frac{\nu+\alpha_2}{2}+k; \rho^2\right).
$$

This concludes the proof. □

**Proof of Proposition 5.** Evaluating the density (6) in the definition (16), we have

$$
\mathcal{H}(\mathbf{Y}) = -\int_0^\infty \int_0^\infty f_{\mathbf{Y}}(\mathbf{y}) \log \left\{ \frac{\left(\frac{\beta}{2}\right)^{\nu+\frac{\alpha_1+\alpha_2}{2}} y_1^{(\nu+\alpha_1)/2-1} y_2^{(\nu+\alpha_2)/2-1} e^{-\frac{\beta}{2(1-\rho^2)}(y_1+y_2)}}{\Gamma\left(\frac{\nu+\alpha_1}{2}\right)\Gamma\left(\frac{\nu+\alpha_2}{2}\right)(1-\rho^2)^{\nu/2}} \right\} dy_1 dy_2
$$

$$
- \int_0^\infty \int_0^\infty f_{\mathbf{Y}}(\mathbf{y}) \log \left\{ \sum_{k=0}^\infty \frac{\left(\frac{\nu}{2}\right)_k}{k!\left(\frac{\nu+\alpha_1}{2}\right)_k \left(\frac{\nu+\alpha_2}{2}\right)_k} \left( \frac{\beta^2\rho^2 y_1 y_2}{4(1-\rho^2)^2} \right)^k \right.
$$

$$
\left. \times F_{1,1}\left(\frac{\alpha_1}{2};\frac{\nu+\alpha_1}{2}+k;\frac{\beta\rho^2 y_1}{2(1-\rho^2)}\right) F_{1,1}\left(\frac{\alpha_2}{2};\frac{\nu+\alpha_2}{2}+k;\frac{\beta\rho^2 y_2}{2(1-\rho^2)}\right) \right\} dy_1 dy_2.
$$

Then

$$
\mathcal{H}(\mathbf{Y}) = -\log \left\{ \frac{\left(\frac{\beta}{2}\right)^{\nu+\frac{\alpha_1+\alpha_2}{2}}(1-\rho^2)^{-\nu/2}}{\Gamma\left(\frac{\nu+\alpha_1}{2}\right)\Gamma\left(\frac{\nu+\alpha_2}{2}\right)} \right\} + \frac{\beta}{2(1-\rho^2)}(\mathbb{E}_{\mathbf{Y}}[Y_1]+\mathbb{E}_{\mathbf{Y}}[Y_2])
$$

$$
- \left(\frac{\nu+\alpha_1}{2}-1\right)\mathbb{E}_{\mathbf{Y}}[\log Y_1] - \left(\frac{\nu+\alpha_2}{2}-1\right)\mathbb{E}_{\mathbf{Y}}[\log Y_2]
$$

$$
- \int_0^\infty \int_0^\infty f_{\mathbf{Y}}(\mathbf{y}) \log \left\{ \sum_{k=0}^\infty \frac{\left(\frac{\nu}{2}\right)_k}{k!\left(\frac{\nu+\alpha_1}{2}\right)_k \left(\frac{\nu+\alpha_2}{2}\right)_k} \left( \frac{\beta^2\rho^2 y_1 y_2}{4(1-\rho^2)^2} \right)^k \right.
$$

$$
\left. \times F_{1,1}\left(\frac{\alpha_1}{2};\frac{\nu+\alpha_1}{2}+k;\frac{\beta\rho^2 y_1}{2(1-\rho^2)}\right) F_{1,1}\left(\frac{\alpha_2}{2};\frac{\nu+\alpha_2}{2}+k;\frac{\beta\rho^2 y_2}{2(1-\rho^2)}\right) \right\} dy_1 dy_2.
$$

Assuming in the pdf (6) that its sum converges at $k=0$ (first term), the differential entropy of $\mathbf{Y}$ can be approximated by

$$
\mathcal{H}(\mathbf{Y}) \approx -\log \left\{ \frac{\left(\frac{\beta}{2}\right)^{\nu+\frac{\alpha_1+\alpha_2}{2}}(1-\rho^2)^{-\nu/2}}{\Gamma\left(\frac{\nu+\alpha_1}{2}\right)\Gamma\left(\frac{\nu+\alpha_2}{2}\right)} \right\} + \frac{\beta}{2(1-\rho^2)}(\mathbb{E}_{\mathbf{Y}}[Y_1]+\mathbb{E}_{\mathbf{Y}}[Y_2])
$$

$$
- \left(\frac{\nu+\alpha_1}{2}-1\right)\mathbb{E}_{\mathbf{Y}}[\log Y_1] - \left(\frac{\nu+\alpha_2}{2}-1\right)\mathbb{E}_{\mathbf{Y}}[\log Y_2] \tag{A15}
$$

$$
- \mathbb{E}_{\mathbf{Y}}\left[\log\left\{F_{1,1}\left(\frac{\alpha_1}{2};\frac{\nu+\alpha_1}{2};\frac{\beta\rho^2 Y_1}{2(1-\rho^2)}\right)\right\}\right] - \mathbb{E}_{\mathbf{Y}}\left[\log\left\{F_{1,1}\left(\frac{\alpha_2}{2};\frac{\nu+\alpha_2}{2};\frac{\beta\rho^2 Y_2}{2(1-\rho^2)}\right)\right\}\right].
$$

Considering $n=2$ in Remark 1, we obtain

$$
F_{1,1}\left(\frac{\alpha_k}{2};\frac{\nu+\alpha_k}{2};\frac{\beta\rho^2 y_k}{2(1-\rho^2)}\right) \approx 1 + \frac{\left(\frac{\alpha_k}{2}\right)_1}{\left(\frac{\nu+\alpha_k}{2}\right)_1}\left(\frac{\beta\rho^2 y_k}{2(1-\rho^2)}\right), \quad k=1,2.
$$

Therefore, using Remark 2, the expected values of (A15) can be approximated by

$$
\mathbb{E}_{\mathbf{Y}}\left[\log\left\{F_{1,1}\left(\frac{\alpha_k}{2};\frac{\nu+\alpha_k}{2};\frac{\beta\rho^2 Y_k}{2(1-\rho^2)}\right)\right\}\right] \approx \frac{\left(\frac{\alpha_k}{2}\right)_1}{\left(\frac{\nu+\alpha_k}{2}\right)_1}\left(\frac{\beta\rho^2}{2(1-\rho^2)}\right)\mathbb{E}_{\mathbf{Y}}[Y_k], \quad k=1,2. \tag{A16}
$$

This concludes the proof. □

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
