# Peer review of "A Generalization of the Bivariate Gamma Distribution Based on Generalized Hypergeometric Functions"

_mathematics, doi:10.3390/math10091502_

Round 1

Reviewer 1 Report

Report on mathematics- 1666412

Title: A Generalization of the Bivariate Gamma distribution based on Generalized Hypergeometric functions

  1. Brief summary of the content of the manuscript

The authors propose a generalization of the bivariate Gamma distribution of Kibble’s type. The distributional properties are presented : pdf, cdf, moments, characteristics functions, hazard function, entropy, mutual information. Numerical simulations are carried out.

  1. Reasoning behind my recommendation

Interesting distributional results are presented : probability and cumulative distribution functions, moment generation and characteristic functions, correlation and cross-product, moment, hazard, Bonferroni and Lorenz functions, an approximation for the Shannon entropy and mutual information index.

However, there are too many formulas in the main text, and some of them should be put in appendices to make the reading easier.

  1. Provide more lists of your minor for the improvement of the manuscript.

“Hazzard” should be replaced with “Hazard” throughout the document.

  • Page 1: the first sentence of the introduction should be rewritten as U1 and U2 are not defined.
  • Page 1: instead of the notation pFq , the more conventional Fpq should be used.
  • Page 1, equation (2) : the right member is ambiguous with commas appearing at wrong places (between factors in a product)
  • Page 2, line 50 : underline that the distribution of W is defined by equation (3)
  • Page 4, theorem 2 : the joint “cdf “ instead of “joint pdf”.
  • Page 14, Figure 3 : the notation MII is not visible over the bars.
  • Page 14, line 132 : a reference is missing
  • Lines and 16, 40 and 149 : “Numerical simulations showed the behavior and performance of exact and approximated expressions, respectively.” This ambiguous sentence is used 3 times and should be reformulated.

Author Response

Please see attached file below

Reviewer 2 Report

Generally, the paper is well written and sounds mathematically. Bi-variate distributions have many potential applications, so the article should be interesting for readers: scientists and engineers. I suggest the Authors a detailed proofreading of the manuscript while preparing a final version of the paper.

Author Response

Please see attached file below

Reviewer 3 Report

Summing up, the authors of the current manuscript proposes a new bivariate distribution obtained from a Kibble-type bivariate
gamma distribution. From the proposed procedure,  the resulting density considers an infinite series of products of two confluent hypergeometric functions. The manuscript is well organized and it represents an interesting contribution in its field. 

However, it is worth mentioning that some authors have recently proposed a procedure called the Probability Transformation method which, based on the theory of the space transformation of random variables and on the principle of probability conservation, is able to evaluate the probability density function of n-dimensional invertible transformation. Some questions: 

- Is the derivation of the proposed bivariate Gamma distribution based on the above theorem? If this is the case, that theorem and the related recent literature should be cited. If conversely, the authors should be shown the advantages of the application of the proposed formulas. 

-In the abstract and in the conclusion section is reported (cite): “Numerical simulations show the behavior and performance of exact and approximated expressions, respectively”. In the reviewer's opinion, a reader would like to know more details. Also, the numerical simulation and its figures do not provide main help along this direction. 

-In the reviewer's opinion, a reader would like to know more details about the simulation of the generalized hypergeometric function. Therefore, it is useful to mention the software used in numerical simulation experiments. 

Author Response

Please see attached file below
